

# Changes in tropical cyclone-associated precipitation of highly damaging Philippine typhoons using high-resolution PGW simulations and multiple-experiment approach

Rafaela Jane Delfino[1,2,3*]; Gerry Bagtasa[1,3]; Pier Luigi Vidale[2,4]; Kevin Hodges[2,4]

[1] Institute of Environmental Science & Meteorology, University of the Philippines - Diliman, Quezon City, Philippines

[2] Department of Meteorology, University of Reading, Reading, United Kingdom

[3] Natural Science Research Institute, University of the Philippines - Diliman, Quezon City, Philippines

[4] National Center for Atmospheric Sciences, Reading, United Kingdom

* *Correspondence to*: Rafala Jane Delfino (rdelfino@iesm.upd.edu.ph)

ORCID ID: Rafaela Jane Delfino 0000-0001-8612-0342; Pier Luigi Vidale 0000-0002-1800-8460; Gerry Bagtasa 0000-0002-5433-7122

**Abstract.** This study has investigated the changes in tropical cyclone (TC)-associated precipitation in the Philippines under past (pre-industrial) and future climate scenarios using the pseudo-global warming technique and dynamical downscaling. What is novel in this work is the use of high-resolution PGW simulations (3km and 5km) and a multiple-experiment approach to directly quantify TC precipitation changes over the Philippines, revealing the nonlinear response of precipitation scaling to different warming pathways. Future climate simulations project a significant increase in TC precipitation, consistent with Clausius-Clapeyron (CC) scaling expectations. However, small deviations from this expected scaling are noted, attributed to factors such as increased TC intensity and atmospheric warming. The simulated TC precipitation in the past climate is found to be lower than that in the current climate. Our convection-permitting model experiments estimate that the average TC inner-core precipitation rate changes from past to current climate conditions are 6% and 8% for the 5km and 3km, respectively. Under the SSP5-8.5 future scenario, simulations indicate a robust rise (by approximately 6% per 1K increase in SST relative to the current climate) in the mean precipitation rates for intense TCs such as Haiyan (2013), Bopha (2012), and Mangkhut (2018) in both the 5km and 3km experiments. Notably, simulations that warm only land and sea surfaces show increases exceeding CC expectations, reaching up to 13% per 1K increase in SST. Increases in both radial and vertical extent of rain are observed. Our analysis shows that these changes are linked to enhanced latent heating, moisture, and updrafts in the TCs' inner-core regions, emphasizing intricate interactions between atmospheric processes and the evolving structure of TCs. Our study underscores that variations in TC intensity and structure play a crucial role in influencing the scaling relationship between sea surface temperatures and TC-associated precipitation in the Philippines.

Keywords: tropical cyclones, precipitation, Philippines, global warming, Clausius-Clapeyron scaling





## 1. Introduction

Tropical cyclones (TCs) is a major source of rainfall and freshwater resources in the Philippines (Bagtasa, 2017; 2022; Yumul *et al*., 2012). In some regions, over half of the annual rainfall is associated with TC-induced precipitation (Bagtasa, 2017; Kubota and Wang, 2009). Projected warming is expected to intensify tropical cyclone rainfall, raising risks of flooding and landslides (Knutson *et al*., 2021; Liu *et al*., 2019; Kossin, 2018).

A common reference is the Clausius–Clapeyron (CC) scaling, which suggests about 7% more atmospheric moisture per degree of warming, enhancing rainfall potential. (Trenberth et al., 2007; Allen & Ingram, 2002; Held & Soden, 2006). The IPCC AR6 concludes that average TC rainfall rates are very likely to increase with continued warming, and that peak rainfall rates may surpass CCS scaling in certain regions. A multi-model study by Knutson et al. (2020) projected a global increase in TC rainfall of +6% to +22% under a +2°C warming scenario. In the Western North Pacific (WNP), reported increases consistently fall within +5% to +7% per °C (Wang et al., 2014; 2015). Similarly, the ESCAP/WMO Typhoon Committee assessment (Cha et al., 2020) estimated a median rise of 17% in TC precipitation rates for the WNP, with a 10th–90th percentile range spanning +6% to +24%.However, TC-associated precipitation may increase at higher rates per degree change in temperature due to dynamic processes, such as increased latent heat fluxes and stronger convergence, which amplify rainfall beyond the thermodynamic expectations (Shi et al., 2024). Empirical studies have supported the application of CCS in explaining increases in TC precipitation (Trenberth et al., 2007; Vecchi et al., 2008). Nonetheless, regional discrepancies have been observed. Kossin et al. (2017) and O'Gorman (2020) noted that observed precipitation changes in certain regions deviate from CCS, likely due to TC dynamics, moisture transport, and local atmospheric circulation.

Recent literature describes Super-CCS, where rainfall rises faster than the 7% per °C benchmark. For example, Liu et al. (2019) and Huprikar et al. (2024) documented inner-core rainfall increases beyond CC expectations, linked to eyewall expansion and stronger updrafts. Liu et al. (2019) reported that rainfall rates within a 100-km radius from the center for TCs with tropical storm intensity could increase by 13%–17% per °C under 21st-century warming. Huprikar et al. (2024) also found that future rainfall associated with Hurricane Irma exceeded CCS expectations. These findings suggest that TC inner-core intensification and structural changes, such as vertical expansion of the eyewall and enhanced updraft, play a critical role in amplifying precipitation beyond the CCS rate.

Despite these findings, there is still a limited understanding of whether TC-associated rainfall in the Philippines conforms to CCS or exhibits Super-CCS properties, especially under changing climate conditions where TCs are projected to intensify (Delfino et al., 2023; 2024). While previous studies have examined TC intensity and precipitation changes (Villarini et al., 2014; Patricola & Wehner, 2018; Liu et al., 2019; Xi et al., 2023), detailed analyses that isolate rainfall scaling with warming remain sparse. Unlike previous case-specific studies, this work systematically applies the PGW framework to multiple Philippine landfalling TCs across three climate states (pre-industrial, present, and future). This study hopes to provide additional evidence of how warming will intensify landfall rainfall hazards in the Philippines. Specifically, we investigate the following:





- How does the TC-associated precipitation in the Philippines change under past and future climate scenarios, and to what extent do these changes align with the expectations of CCS?
- How do variations in TC intensity and structure influence the scaling relationship between sea surface temperatures and TC-associated precipitation in the Philippines?

A better understanding of these mechanisms is critical for anticipating current and future flood risks and improving disaster preparedness in the Philippines. Section 2 of this paper discusses the methods, Section 3 provides the results and discussion, and Section 4 highlights the conclusions.

## 2. Methods

### 2.1 Model Configuration

We used WRF-ARW v3.8.1 (Skamarock et al., 2008) with two main setups: (i) a nested 25 km–5 km grid with cumulus parameterization and (ii) a convection-permitting single 3 km grid without cumulus parameterization. Both applied 44 vertical levels from surface to 50 hPa. Further physics choices and TC tracking follow Delfino et al. (2023).

### 2.2 Experimental Design

Three TC cases are selected based on the region in the Philippines where the TCs made landfall, the month of occurrence, and associated damages – Typhoons Haiyan (2013), Bopha (2012), and Mangkut (2018). More detailed information on these three TC cases is described in Delfino et al. (2023). The three TC cases were simulated with four different initialization times (00, 06, 12, and 18 UTC) to create an ensemble from a single driving reanalysis, thereby minimizing uncertainties from variations in the initial conditions. For tracking the simulated TCs, the simulated track and intensity values were obtained every 6 hours using the TRACK algorithm (Hodges et al., 2017) as used in Hodges and Klingaman (2019) and Delfino et al. (2023).

Multi-model datasets from the Coupled Model Intercomparison Project phase 6 (CMIP6) were used to apply the Pseudo-Global Warming (PGW) method. We simulated the three TC cases under different climate conditions by primarily adjusting the following parameters: SST, atmospheric temperature, and relative humidity (RH) between current, pre-industrial, and future climate conditions. Four CMIP6 models were used, and results for all ensemble members were averaged for each of the CMIP6 models. The models - HadGEM3-C31-LL, CESM2, MIROC6, MPI-ESM1-2-HR – were chosen to represent different warming levels, under the Shared Socio-economic Pathways (SSP) 5-8.5 scenario. Monthly mean deltas from CMIP6 (2070–2099 minus historical) were applied to ERA5 boundary and initial fields. Three experiment sets were tested: (1) SST-only (SFC), (2) SST plus tropospheric temperature (SFC+PLEV), and (3) SST, temperature, and humidity (FULL). Each case was simulated with four initialization times to form ensembles. A more detailed description of the methodology can be found in Delfino et al. (2023).



It is also important to note here that the PGW technique faces challenges related to spin-up and dynamical balance
when simulating TCs. Spin-up issues arise because the initial conditions need time to adjust to imposed future
climate anomalies, potentially leading to unrealistic results during the adjustment period. Additionally, altering
these conditions can disrupt the dynamic balance of the atmospheric system, resulting in inaccuracies in the
intensity and behavior of simulated TCs. These issues were explored and addressed in Delfino et al., (2024).
**2.3 TC precipitation analysis**
Rainfall analysis included hourly rates, accumulated totals, and reflectivity composites during peak intensity. The
forward sector of each storm was emphasized, as this region typically hosts the strongest convection due to storm
motion combined with cyclonic rotation. We also investigated the relationships between TC rain rate and TC wind
speed, emphasizing how this relationship varies within the TC inner-core region (2.5˚ distance from the center).
We also analysed the simulated reflectivity rates (in dbz) from the WRF simulations and vertical profiles of the
averaged composites over multiple time steps (e.g., average reflectivity during peak intensity hours) of simulated
reflectivity over a 2.5˚ radius in the forward direction of the storms for the simulations, since TCs typically have
the highest precipitation rates and strongest reflectivity in their forward quadrant due to the combined effects of
TC motion and cyclonic rotation. By focusing on this region, we can capture the most intense precipitation features
and better understand the TC's impact. Reflectivity is particularly useful because it provides insight into the
microphysical structure of convection within TCs. Higher reflectivity values typically correspond to deeper and
more intense convective cores, which are closely linked to strong updrafts and heavy rainfall production. The
profiling is done at peak intensity at height levels between 0 and 18 km, and the radial grid extends to 10° for each
simulation.
**3.   Results and Discussion**
**3.1 Changes in the total accumulated rainfall**
The discussions in Sections 3.1 and 3.2 below are from the FULL experiments, with additional discussions on
other sets of experiments in Section 3.3. In the FULL experiments, rainfall consistently increased under future
warming compared with the present, while simulations under pre-industrial conditions produced less precipitation.
This progression highlights stronger rainfall across pre-industrial, present-day, and future climates. Future
precipitation changes in the FULL experiments reached a maximum percentage change in hourly rainfall rate of
18% (18%) for Haiyan, 11% (20%) for Bopha, and 26% (7%) for Mangkhut in the 5kmCU (3kmNoCU) runs
under future climate conditions; and up to more than 18% (18%) in total accumulated precipitation over the
simulation period for Typhoon Haiyan, 12% (20%) for Bopha, and 25% (7%) for Mangkhut (Figure 1).  In
contrast, in the simulations under pre-industrial climate conditions, TC-associated precipitation are less than the
current climate, with percentage changes (pre-industrial minus current) in total accumulated precipitation over the
simulation period of -3%(-2%) for Haiyan, -20%(0%) for Bopha, and -6%(-16%) for Mangkhut; and percentage





change in precipitation rate of -3%(-2%) for Haiyan, -19%(0%) for Bopha, and -2%(-16%) for Mangkhut in the
5kmCU(3kmNoCU) runs (Figure 1).

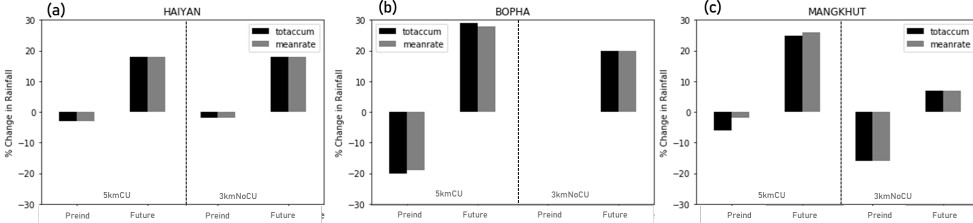


*Figure 1. Percent changes in total accumulated and mean precipitation rate relative to current climate for the*
*5kmCU runs (left panel) and the 3kmNoCU runs (right panel) under the pre-industrial and future climate*
*relative to current climate conditions for (a) Typhoon Haiyan, (b) Typhoon Bopha, and (c) Typhoon Mangkhut*

Examination of the time series of accumulated precipitation, together with variations in RH, shows that
precipitation under the future climate generally exceeds that of the current climate, while the current climate also
produces greater TC rainfall than the pre-industrial period across both variables (Figure 2). Under future climate
conditions, accumulated precipitation consistently and markedly increases relative to the current climate,
suggesting the likelihood of more intense and prolonged precipitation episodes. . Similarly, the current climate
model runs exhibit higher accumulated precipitation than those of the pre-industrial period, reflecting an ongoing
trend of changing precipitation patterns over time. Simultaneously, changes in RH are scrutinized in conjunction
with the accumulated precipitation. The analysis reveals the same upward trajectory, with higher RH levels
(Figure 2, right panels) in both the current and future climates compared to the pre-industrial era.



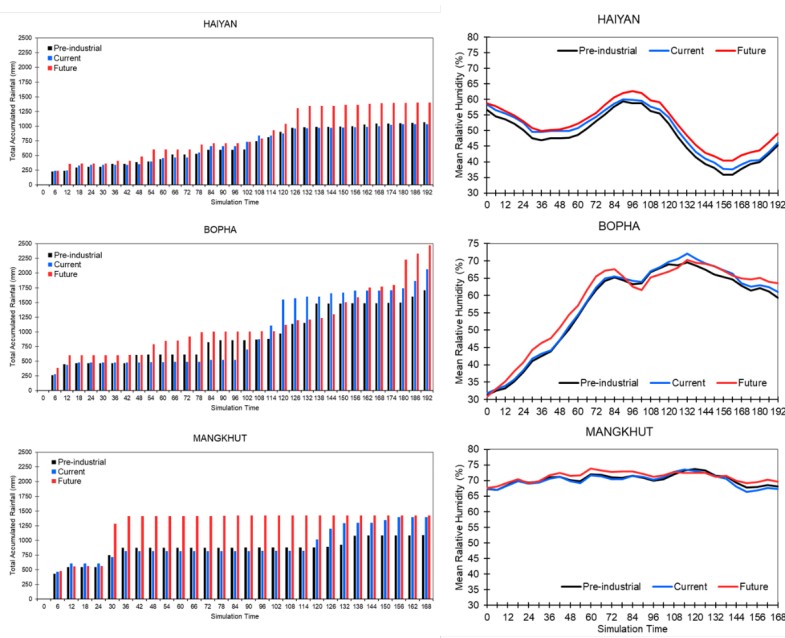

***Figure 2. Total accumulated precipitation (left panel) and mean mid-tropospheric RH (right panel) of the TC cases under pre-industrial, current, and future climate throughout the simulation period of the three TC cases under the 5kmCU simulations.***

Figure 3 presents the differences in total accumulated rainfall between Current minus Pre-industrial (upper panels) and Future minus Current (lower panels). In Figure 3a, a statistically significant overall increase (p = 0.0066) is evident, particularly across the central Philippines. For Bopha, moderate increases are observed over parts of Mindanao and the Visayas, with a mean significant rise of 1.7%. Mangkhut, by contrast, displays a pronounced increase in total accumulated rainfall, most notably over northern Luzon, with a mean rise of 14.2 mm (10.5%), highly significant at p < 0.0001. In the Future minus Current climate comparisons, Haiyan shows a marked increase in rainfall across the Visayas and central Mindanao, with a mean increase of 10.1 mm (7.4%), also statistically significant. Bopha shows the largest increase, especially over Luzon and the Visayas, with an average rise of 26.3 mm (24.3%). Mangkhut, however, exhibits only a minimal increase in rainfall over northern Luzon and adjacent regions, with a mean increase of 3.9%, largely attributable to a slight northward displacement of its track in the future climate simulations.

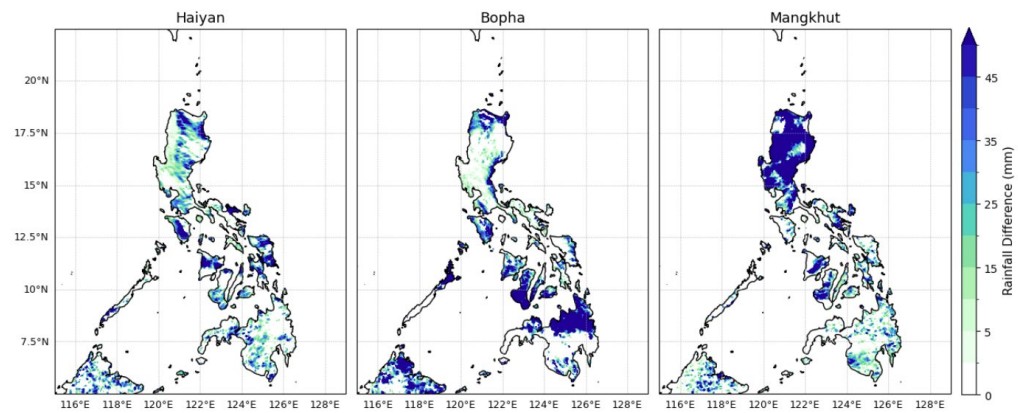

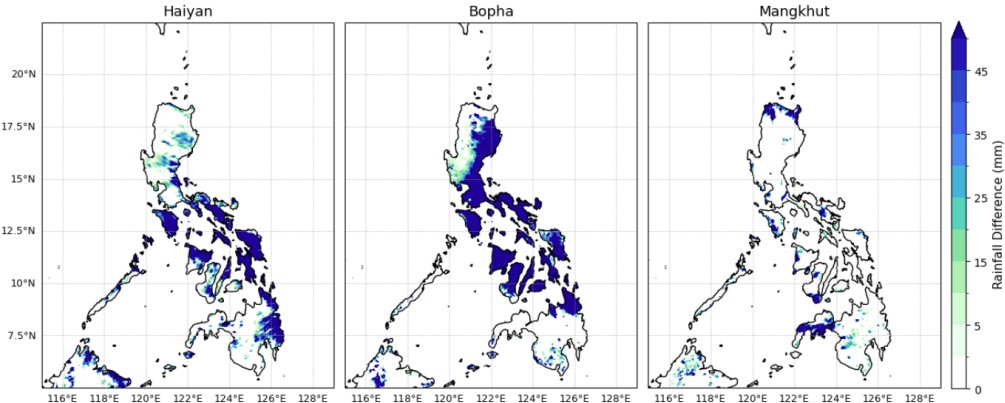

**Figure 3. Difference in the ensemble-mean accumulated precipitation (a-c, top panels) Current minus Pre-industrial; (d-f, bottom panels) Future minus Current for Typhoons Haiyan (left), Bopha (middle) and Mangkhut (right) under the 3kmNoCU simulations.**

Figure 4 shows the boxplots of the distribution of total accumulated rainfall (in mm) for the three typhoons under the three climate scenarios: Preindustrial (black), Current (blue), and Future (red). This shows that both Haiyan and Bopha exhibit substantial increases in total rainfall with warming, whereas Mangkhut shows a relatively smaller increase. For Typhoon Haiyan, the ensemble median total rainfall increases in the Future scenario by approximately 9% increase. For Bopha, we are looking at approximately 23% increase from preindustrial to the future scenario, while Mangkhut exhibits minimal change. The interquartile range (IQR) also shifts upward, with higher ensemble spread under future warming for all three TCs, again with relatively minimal change for Mangkhut compared to Haiyan and Bopha. The current scenario shows little deviation from preindustrial.





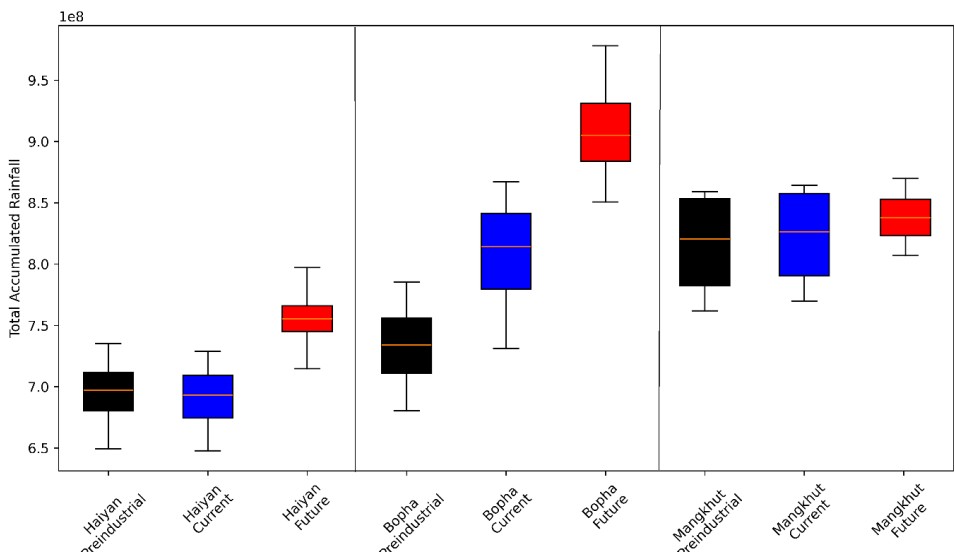

206

**Figure 4. Boxplots for the Total Accumulated Rainfall within a 2.5° radius of the center of the track from**
**the preindustrial, current, and future climate scenarios for Typhoons Haiyan, Bopha, and Mangkhut under**
**the 3kmNoCU simulations using all ensemble members.**

210

*3.2 Changes in TC rain rate and intensity in the simulations*

212

Figure 5 below shows the frequency distributions of the rainfall rates for Haiyan, Bopha, and Mangkhut across
three climate scenarios (Preindustrial, Current, Future), with the left panels showing the frequency of different
precipitation rates (in mm/day) and the right panels showing the distributions of the rainfall amount associated
with each precipitation rate. Based on this figure, the future scenarios (red line) show a rightward extension (tail),
indicating an increase in the frequency of higher rates across all TCs. Meanwhile, most total rainfall amounts
(right panels) in current and future climates are in the higher end (≥100 mm/day).

219

220



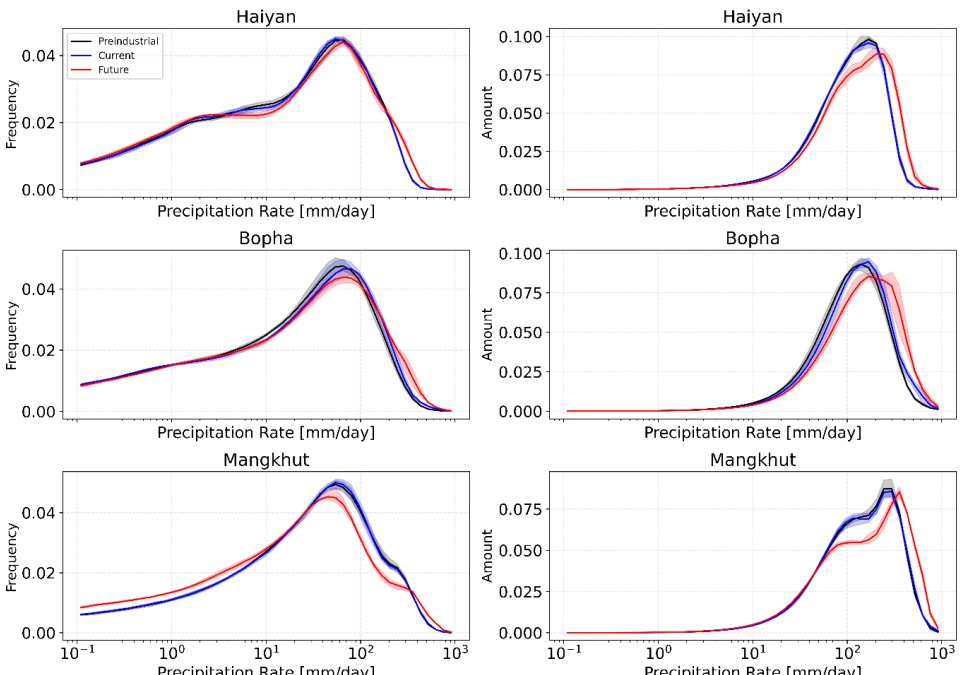

**Figure 5. Distributions of Typhoons Haiyan, Bopha, and Mangkhut's precipitation rate (left panels), frequencies (%), and (right panels) amounts (mm d-1) for the pre-industrial, current, and future scenarios under the 3kmNoCU simulations using all ensemble members.**

Figure 6 shows the boxplots of percent changes in mean, 95th, and 99th percentiles of 6-hourly rainfall rates for Haiyan, Bopha, and Mangkhut relative to the current climate scenario. Future simulations (red boxes) show consistently higher median percent increases, particularly for the mean rainfall across all TCs. However, the rainfall extremes (95th and 99th percentiles) increase more modestly, but in most cases, they still exceed the 7%/K baseline, suggesting potential influence from TC intensity. The figure also shows a high spread in the ensemble in terms of spread and outliers, particularly in the percent changes in mean rainfall.



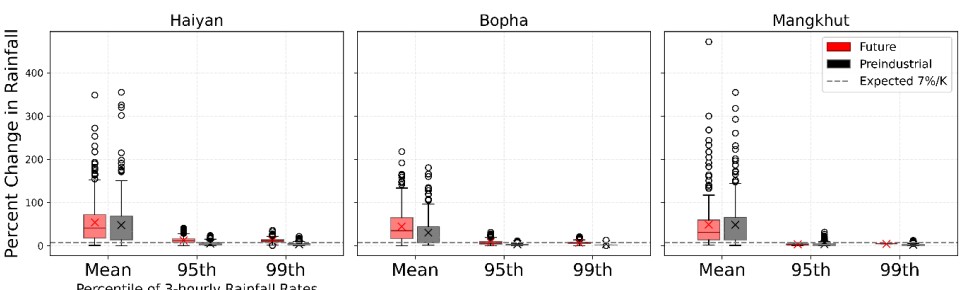

**Figure 6. Percent change in 6-Hourly Rainfall Rates of the Pre-industrial and Future simulations compared to Current simulations for Typhoons Haiyan, Bopha, and Mangkhut. The x-axis shows different metrics: mean 6-hourly rate, and 95th and 99th percentiles. The X markers denote the average percent change, and the black**

A recent study by Macalalad et al. (2023) investigated the effects of historical warming on a different TC case, Typhoon Vamco (2020), and found that the influence of historical warming are counteracted by additional factors such as orography and topography, resulting in comparable precipitation levels between past and present simulations within two river basins in the Philippines. To isolate the topographic effects of TC-associated precipitation, we analysed the TC precipitation rate at the time the simulated TCs reached peak intensity prior to landfall. In the 3kmNoCU experiments, the percent change in average rain rate for Future minus Current (and Current minus Pre-industrial) is 10% (1%) for Typhoon Haiyan, 13% (10%) for Typhoon Bopha, and 9% (14%) for Typhoon Mangkhut. In the 5kmCU experiments, the corresponding percent changes are 17% (1%) for Haiyan, 11% (13%) for Bopha, and 14% (5%) for Mangkhut. Increases in precipitation are concentrated in the inner-core regions of the TCs (Figures 7, 8, and 9 for Haiyan, Bopha, and Mangkhut, respectively), with the signal more pronounced in the 5kmCU simulations. In the 3kmNoCU simulations, there are coherent spatial patterns in the future precipitation response characterized by drying in the outer core, resulting in precipitation responses that are stronger over the inner core region in all three TC cases. This drying outer core condition is also present in the 5kmCU future run for Mangkhut. Such drying has also been found by Patricola and Wehner (2018), particularly in the weaker TCs.



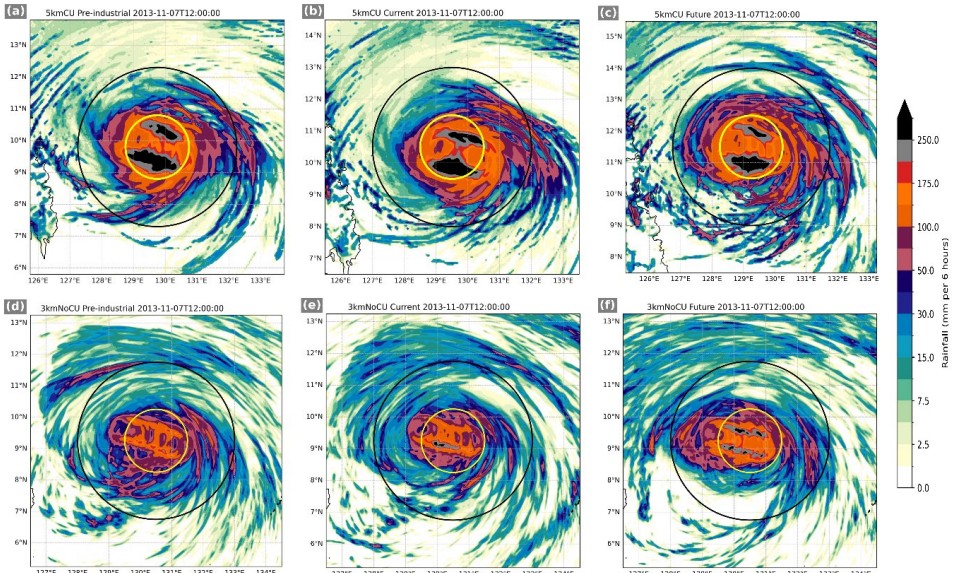

**Figure 7. Simulated precipitation rate (mm/hr) at simulated peak intensity for Haiyan on 7 November 2013 12UTC for the 5kmCU runs (upper panel) and the 3kmNoCU runs (lower panel) under the pre-industrial (left), current (middle), and future climate conditions. The black circle indicates the 2.5° radius from the center, and the yellow circle indicates the 1° radius from the center.**

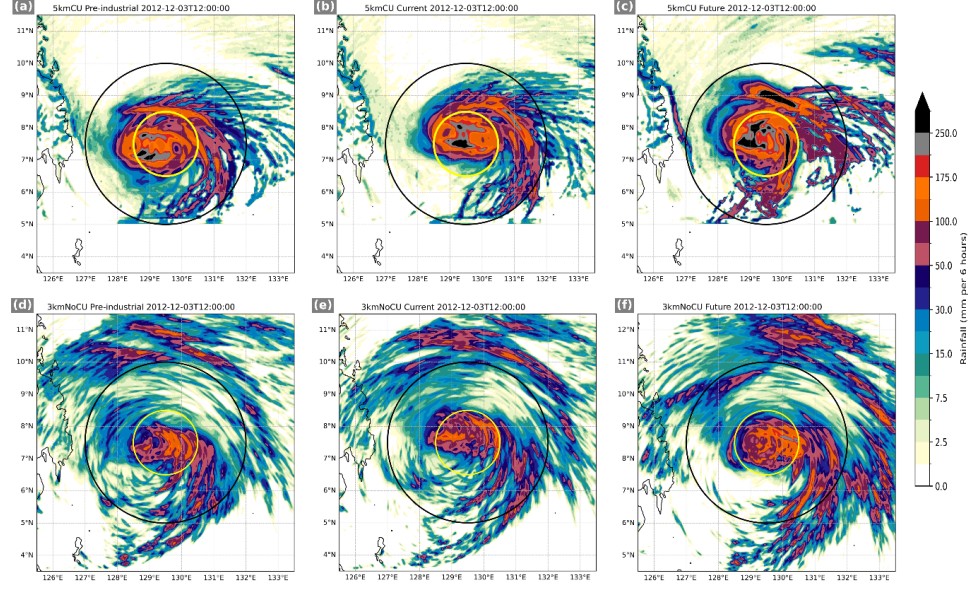

**Figure 8. Same as Figure 7, but for Bopha**



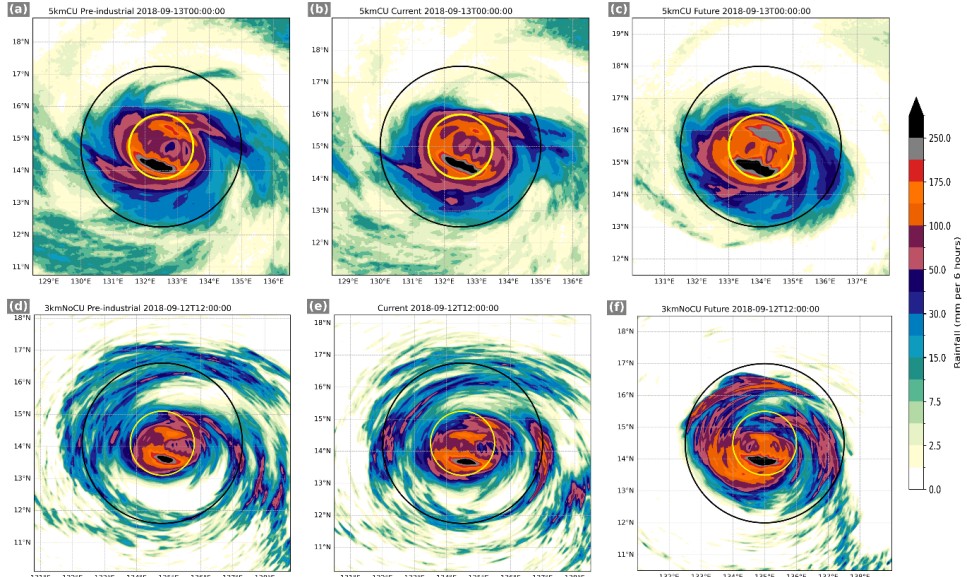

**Figure 9. Same as Figure 7, but for Mangkhut**

### *3.3 Changes in TC rain rate and Clausius-Clapeyron scaling*

According to Liu *et al.* (2019), a projected increase in precipitation rate that is more than what is expected according to the CCS relation may be linked to enhanced TC intensity associated with land surface and SST warming. As shown in Figure 10a, the largest increases in precipitation tend to occur when only the surface temperatures have been warmed (SFC only). In contrast, a lower increase is found when atmospheric warming is also included (SFC+PLEV) and RH changes (FULL). In the SFC-only simulations, future precipitation changes exceed 17% per degree of SST warming, surpassing the CCS rate..

By comparison, both the 5kmCU and 3kmNoCU simulations with varying initializations (Figure 10b) remain within this thermodynamic expectation under future climate conditions. Likewise, the change in rain rate relative to the pre-industrial climate is consistently within the CCS for all typhoons (not shown). These results are consistent with Stansfield and Reed (2023), who found that the apparent scaling of TC precipitation in response to SST warming is typically around 6–9% per K, consistent with the CCS rate, while the climate scaling - which accounts for long-term climate changes - is smaller, around 5% per K. They emphasized that the apparent scaling reflects short-term changes driven by SST alone, while climate scaling incorporates broader atmospheric changes like shifts in wind shear, which reduce the precipitation intensification seen in the SFC-only experiments of Liu et al. (2019).



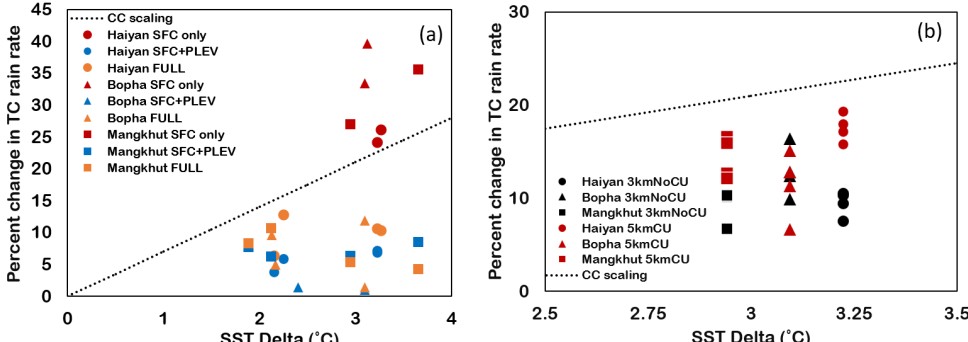

**Figure 10. Percentage change in TC inner core rain rates vis-à-vis SST delta. (a) 5kmCU simulations using different levels of PGW delta and (b) 5kmCU and 3kmNoCU simulations using different initializations. The black dotted lines show the Clausius-Clapeyron scaling.**

Figure 11 shows the relationship between simulated TC intensity and inner-core precipitation rates from the 5kmCU and 3kmNoCU experiments. In the present-climate simulations, TC intensity and inner-core rain rate exhibit an approximately linear relationship, with a regression slope of 0.807 (mm h$^{-1}$) per (m s$^{-1}$) (Figure 11a). Under future and pre-industrial climate conditions, both the 5kmCU and 3kmNoCU experiments yield a similar relationship, though with a lower regression slope of 0.38 (mm h$^{-1}$) per (m s$^{-1}$) (Figure 11b). Overall, inner-core rain rate is positively correlated with TC intensity: in the 5kmCU simulations, a 1 m s$^{-1}$ increase in TC intensity corresponds to a ~1.17 mm h$^{-1}$ increase in rain rate, while in the 3kmNoCU simulations the increase averages ~0.66 mm h$^{-1}$. Considering the sensitivity of TC intensity to SST and the dependence of rain rate on intensity, we further examined the sensitivity of TC precipitation to SST. Our findings indicate that the TC inner-core rain rate of the three TC cases (for both 3kmNoCU and 5kmCU runs) will increase by approximately 6% per 1 K increase in SST in the future. This increase can be understood through the combined effects of TC intensity sensitivity to SST and CCS. However, larger increases (up to 33% per 1K increase in SST) can be found in the 5kmCU SFC-only experiments, which may be due to the changes in thermodynamics and shifts in TC trajectory (Delfino et al., 2023). The rest of the simulations under the 5kmCU SFC+PLEV and FULL and 3kmNoCU experiments show no significant changes in track.



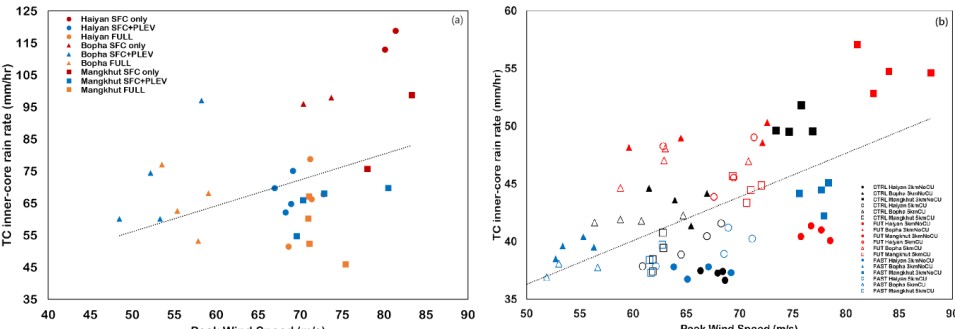


**Figure 11. Dependence of TC inner-core rain rate on TC intensity. (a) 5kmCU simulations using different**
**levels of PGW delta and (b) 5kmCU and 3kmNoCU simulations using different initializations.**


**3.4 Potential mechanisms driving TC-associated precipitation changes**

Figure 12 presents the vertical profiles of the averaged composites of simulated reflectivity within a 2.5° radius
in the forward direction of the typhoons for the 3kmNoCU simulations, the 5kmCU simulations exhibit similar
patterns (not shown). The composites are taken at peak intensity across different height levels between the surface
up to 18 km. Results indicates substantial increases in simulated reflectivity throughout the vertical column,
particularly within the TC's primary circulation. This in turn, leads to deeper core convection in the future climate
simulations, as seen in the upward extension expansion of the eyewall.
The vertical expansion of the eyewall, the inner region of TC with intense convection and strongest winds
surrounding the eye of the storm, is consistent with what is found in the vertical cross-section of the composite of
azimuthally averaged winds (Delfino et al., 2023). As indicated by the increased reflectivity observed in Figure
12, this provides insights into the evolving structure of TCs under the different climate conditions simulated in
the study. Although the composites are taken at the peak intensity of the TCs, the results reveal substantial
enhancements in reflectivity throughout the vertical extent within a 2.5° radius around the TC's forward direction.
This enhanced reflectivity suggests a greater amount of precipitation within the TC's inner core, which has
implications for its overall dynamics. The explanation for the vertical expansion of the eyewall lies in the
interactions between different atmospheric processes. As TC intensity increases, precipitation rates within the
storm also tend to increase (Alvey et al., 2015), reflecting and reinforcing internal dynamical processes critical to
intensification. Although warm sea surface temperatures, weak vertical wind shear, and high low-level moisture
are recognized as necessary environmental conditions for TC development (DeMaria et al., 2005; Kaplan et al.,
2010, 2015), these factors alone are insufficient to account for observed variations in TC intensity (Hendricks et
al., 2010).This has led to a growing focus on internal TC processes, particularly precipitation and convection, as
key drivers of TC intensification. Latent heat release from enhanced precipitation warms the TC core and
contributes to further pressure falls, strengthening the cyclone (Rotunno & Emanuel, 1987; Pendergrass, 2014;
Yamada, 2017). Observational studies using TRMM and passive microwave data have shown that more intense
TCs are typically associated with broader and more symmetric precipitation coverage, especially in the inner core
(Alvey et al., 2015). Stratiform and moderate-to-deep convective precipitation are particularly linked to rapid



intensification (Tao & Jiang, 2015), suggesting that increasing precipitation is not just a result of intensification,
but a contributor to it. Ruan and Wu (2018) also found that as TCs intensify, they exhibit increased precipitation
and colder high cloud tops, and that widespread very deep convective clouds (IR BT < 208 K) are strong predictors
of future intensity change, particularly rapid intensification (Tierra and Bagtasa 2021).

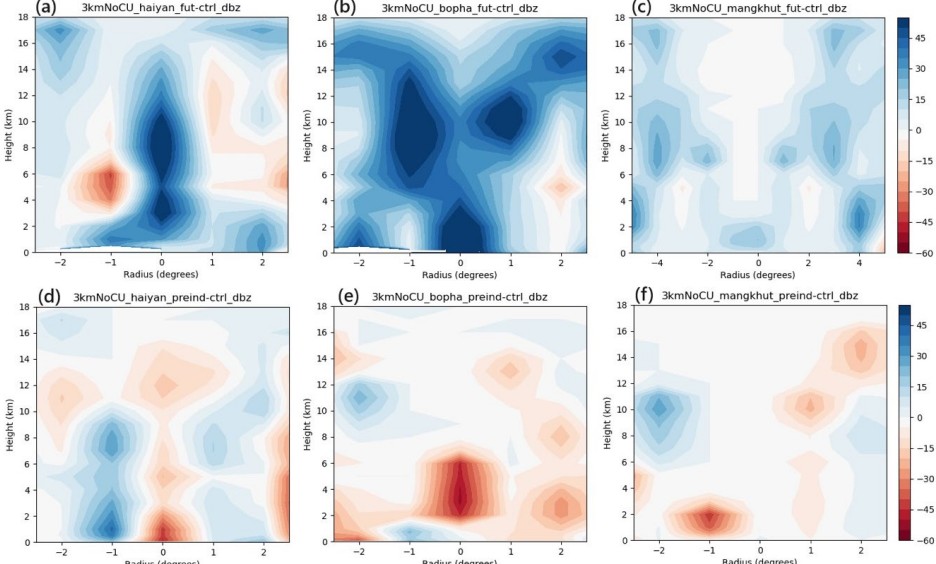


**Figure 12. Radius (in degrees) – height (in km) cross sections of simulated differences in reflectivity (dBZ)**
**from future minus current (top panels, a-c) and pre-industrial minus current (bottom panel, d-f) from the**
**3kmNoCU experiments. All reflectivity fields are at peak intensity while precipitation rates are shown**
**within a 2.5° x 2.5° grid from the center at peak intensity for Typhoon Haiyan (a and d panels), Bopha (b**
**and e panels), and Mangkhut (c and f panels).**


Vertical motion (omega) was investigated to understand the detailed processes of TC-associated precipitation
intensification in different climate conditions (Figure 13). Vertical velocity fields show enhanced ascent near
eyewall regions under warming scenarios, with the largest anomalies in future runs. These dynamical signals are
consistent with reflectivity results, reinforcing the link between stronger updrafts and increased rainfall. Our
analysis revealed significant regions of strong ascending motion, characterized by increased omega, near the
eyewall and along spiral rainbands of Typhoons Haiyan, Bopha, and Mangkhut, primarily within a 250-km radius
from the TC center. These regions are crucial for intense convective activity and high precipitation rates due to
vigorous updrafts and condensation processes. Comparisons between future and current climate simulations reveal
larger differences in omega near the TC core region, indicating a potential increase in vertical motion and
precipitation rates under future climate conditions. Similarly, significant differences are also evident between
current and past climate scenarios, suggesting consistent changes in vertical motion patterns over time. Our
findings regarding the differences in omega between current and future climate simulations suggest a projected
increase in vertical motion and moisture convergence, which will likely lead to stronger precipitation events. In



our earlier work (Delfino et al., 2023), we note that there are some changes or shifts in the vertical profiles of the TC cases that potentially led to the further intensification of rainfall associated with these TCs. Additionally, Shi et al. (2024) emphasized that TC intensification under warming is not driven solely by stronger updrafts, but also by the expansion of deep convective cores accompanied by a suppression of shallow cumulus and congestus clouds. This structural adjustment implies that while localized hourly rainfall may scale with the CCS relationship, precipitation accumulated over broader areas could increase by up to 18% per degree of warming.

Understanding these dynamics is essential for predicting how TCs may evolve under different climate scenarios, particularly in terms of intensity and precipitation distribution. Overall, the study underscores the sensitivity of TC dynamics to changes in vertical motion and provides insights into the complex interactions shaping TC-associated precipitation, contributing to advancing our understanding of TC characteristics in a changing climate.

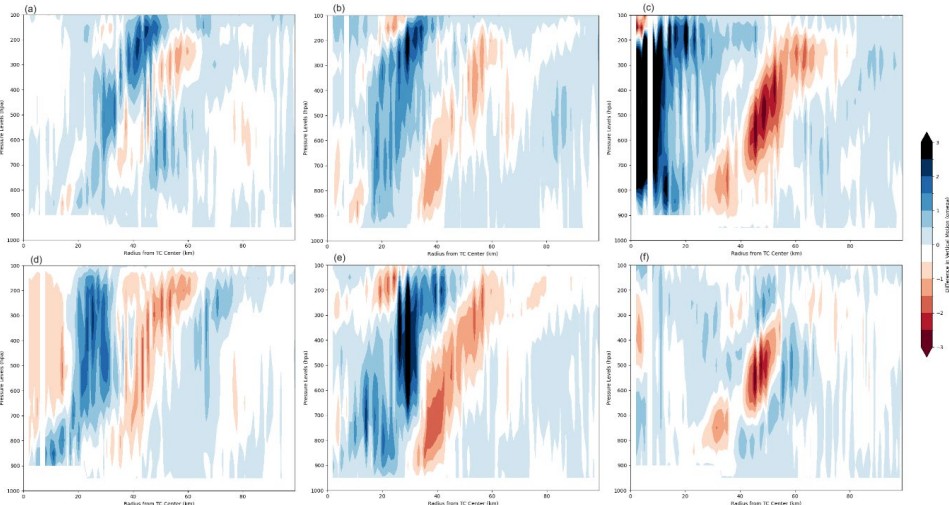

**Figure 13. Simulated difference in vertical velocity, Omega (hPa s−1) at several levels (1000-100hPa) within 250 km radius from the TC center for the 3kmNoCU experiments difference between future and current climate (top, a-c), and difference between current and past climate (bottom, d-f) for Typhoons Haiyan (left), Bopha (center) and Mangkhut (right).**

This work advances beyond our earlier case-specific studies (Delfino et al., 2023; Delfino et al., 2024) by systematically applying the PGW framework to multiple Philippine landfalling cyclones under three climate states (pre-industrial, present, and future). By comparing rainfall scaling across storms, we demonstrate consistent Super-CC behavior in forward quadrants and highlight a progressive intensification of landfall rainfall hazards. These results provide a regional-scale perspective and connect directly to disaster risk in the Philippines, representing a new contribution beyond our previous publications.

## 4. Summary and conclusions



This study investigated the changes in tropical cyclone (TC)-associated precipitation in the Philippines under past
(pre-industrial) and future climate scenarios, using a hierarchy of convection-permitting simulations. Importantly,
this is the first time that such extreme typhoons have been systematically simulated and compared using both
convection-permitting and parameterised convection models. Some of the results show sensitivity to the treatment
of convection, while other experiments are relatively insensitive. Overall, the main findings are robust and largely
insensitive to this model formulation choice. In alignment with the expectations from Clausius–Clapeyron scaling
(CCS) for TCs, the future climate simulations project a robust increase in TC precipitation. Deviations from the
expected CCS scaling are attributed to factors such as increased TC intensity, also driven by atmospheric warming.
Under past climate conditions, TC precipitation in the Philippines would have been generally less than that of
current climate conditions, with an average change (current - past) in TC inner-core precipitation rates of 6% and
8% for the 5kmCU and 3kmNoCU experiments, respectively. The observed changes between the past and current
climate align with expectations from CCS, indicating that the atmosphere holds approximately 7% more water
vapor per degree Celsius increase in surface temperature. Conversely, under future climate scenarios, the FULL
simulations conducted under the SSP5-8.5 scenario indicate a robust rise - by approximately 6% per 1 K increase
in SST in the mean precipitation rates for specific intense TCs, such as Haiyan, Bopha, and Mangkhut, in both the
5kmCU and 3kmNoCU experiments, consistent with CCS expectations.

However, notable deviations from CCS arise in simulations where only the land and sea surface temperatures are
increased (e.g., 5kmCU SFC-only), with precipitation increases reaching up to 13% per 1 K SST increase. This
suggests that additional dynamical processes, such as TC track changes or structural modifications, amplify
precipitation beyond thermodynamic expectations. Frequency distributions of rainfall further confirmed an
intensification of heavy precipitation events in a warming climate. Future scenarios displayed a consistent shift
toward higher 6-hourly rainfall rates, particularly in the right tail (extreme values).

Notably, the results reveal an asymmetry in the response of TC precipitation between past cooling and future
warming. Specifically, under future warming, feedback appears to amplify precipitation (consistent with a positive
feedback mechanism), while in the pre-industrial simulations, these feedback act in the opposite direction,
dampening TC intensity and precipitation.

Spatial rainfall maps show increased precipitation from pre-industrial to present-day, and further into the future.
Haiyan and Bopha display substantial increases, while Mangkhut's response is comparatively modest, reflecting
its more northerly track. Overall, the total accumulated rainfall associated with the three TCs, within a 2.5° radius,
is expected to increase by up to 25% in the future.

Reflectivity profiles reveal broader and taller convective towers in warmer climates, suggesting intensified latent
heating and deeper storm cores, which drive more efficient rainfall production. . This is attributed to enhanced
latent heating, which drives stronger updrafts and contributes to a deeper TC core. Mechanistically, the deeper
cores lead to intensified updrafts that enhance the lift of moist air, promoting additional adiabatic warming within
the TC and further enhancing TC-associated precipitation. This chain of causation aligns with previous studies
(e.g., Yamada et al., 2017) that emphasize the role of latent heating in TC dynamics.




This study extends beyond our earlier analyses by evaluating multiple Philippine landfalling cyclones in a unified
PGW framework across pre-industrial, present, and future climates. The consistent emergence of Super-CC
scaling in forward quadrants underscores a robust intensification of rainfall hazards unique to this region. By
situating these findings within the Philippine disaster risk context, the study provides new insights for impact-
oriented assessments and highlights the need for adaptation planning to address more extreme rainfall in future
tropical cyclones.

Our findings provide critical insights into how variations in TC intensity and structure influence the scaling
relationship between SST and TC-associated precipitation in the Philippines. The heightened intensity of
simulated TCs under future climate conditions contributes to increased TC-associated precipitation rates within
the inner core, diverging from expected CCS behavior. This supports the notion of the CCS, emphasizing the
interplay between atmospheric moistening, TC dynamics, and evolving TC structures.

We recommend that future studies focus on (1) the climatological trends in TC precipitation i.e. effects of climate
change and natural climate variabilities on TC precipitation; (2) the specific impacts of changing TC tracks, the
role of atmospheric moisture distribution, and (3) the influence of varying SST patterns and magnitude on TC-
associated precipitation in the Philippines. Additionally, further research should investigate the potential effects
of land-use changes and urbanization on TC rainfall patterns to provide a more comprehensive understanding of
the local impacts of climate change on TCs.

**AUTHOR DECLARATIONS**

**Funding Information**

RJD was supported by a scholarship from the Philippine Commission on Higher Education and the British Council
through the JDNP Dual PhD Program. KH and PLV were funded by the UK Natural Environment Research
Council (NE/W009587/1). This work used the JASMIN data analysis facility (https://www.jasmin.ac.uk
) and was partly supported by the University of the Philippines Diliman Natural Sciences Research Institute
(NSRI). under the project entitled "*Investigating the effects of past and future changes in climate on a collection
of Philippine Super Typhoons and their flooding potential from the last 10 years (2012 – 2022) using a convection-
permitting regional climate model*" (ESM-24-1-01).

**Author Contributions**
RJD designed the study and prepared the first manuscript draft. All co-authors provided input, interpretation, and
revisions leading to the final paper.
**Conflicts of Interest**
I declare that the authors have no competing interests as defined by Copernicus Publications, or other interests
that might be perceived to influence the results and/or discussion reported in this paper



**Ethics approval/declarations (include appropriate approvals or waivers)**

*Not applicable*

**Consent to participate (include appropriate statements)**

*Not applicable*

**Consent for publication (include appropriate statements)**

*Not applicable*

**Data and Code Availability**

Simulation outputs are hosted on the JASMIN platform. The WRF model (Skamarock et al., 2008) and TRACK tool (Hodges, 1995) were used. Data processing and figures were produced with CF-python and CF-plot and are available from https://www.ncas-cms.github.io/cf-python/.. Code for the WRF model is available at http://www.www2.mmm.ucar.edu/wrf/users/downloads.html. WPS geographical input data are available from https://www.www2.mmm.ucar.edu/wrf/users/download/get_sources_wps_geog.html#mandatory. The codes and simulation data are available upon request to the corresponding author.

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
