# Peer review of "Changes in tropical cyclone-associated precipitation of highly"

_EGUsphere, 2025_

## Referee Comment (RC1)

**1 Overall Recommendation**

The research paper investigates the impact of warming on rainfall changes in damaging Philippine typhoons using high-resolution convection-permitting models with pseudo-global warming methods. It advances the understanding of tropical cyclone (TC) landfall rainfall hazards in the Philippines. However, several improvements are needed before recommending acceptance. I recommend a major revision, with particular attention to: 1) the defining of TC inner core, 2) Quantifying future changes in TC rain rates due to dynamic (e.g., TC intensity increase) and thermodynamic (e.g., water vapor increase) factors, including their relative contributions to inner core rainfall changes (100 km radius). 3) Using a consistent rain rate unit (mm/hour), as the mix of mm/day and mm/6-hour confuses readers.

**2 Major Comment**

The section "Changes in TC rain rate and intensity in the simulations" discuss rain rate change inade-quately. Can you please show the simulation time series of inner core rain rate just like Figure 2 expect shows the rain rate. Can you depose the change of TC rainfall rate to thermodynamic and dynamic contribution to see whether the moisture increase or TC intensity dominate the rainfall change. You can follow Yang and Ralf (2025) (DOI 10.1088/1748-9326/add753).

Figure 11, In the discussion, perform linear regression separately for the Pre-industrial and future climate. Mixing Pre-industrial and future climate together results in no meaningful results. A recent study (Chen et al., 2025, https://doi.org/10.1029/2025GL116146) suggests that in a warming climate, a 1 m s-1 increase in TC intensity corresponds to a higher increase in rain rate because of increased moisture. This result was performed in coarse-resolution GCM. Please explore whether this is valid in your simulations.

Rain rate missmatch exist broadly in your article, like Figure 6, 7, 8, 9. I suspect whether the rain rates in Figure 7, 8, 9 are plottely correctly. It seems there are moving TC center in your Figure 7, 8, 9. The preciptation rate is calculated using difference in 6 hour accumulated rainfall? Probably using difference in 1 hour accumulated rainfall by increasing model output frequency. The 6 hourly accumulated rain difference may falsely enlarge the intense rainfall region due to movement of TC itself. Please use hourly accumulated rain difference. With hourly accumulated rain difference, a compacted intense rainfall (100 km radius) is likely to be found. And 2.5 degree is too large for defining inner core.

Lots of experiments have been done in this study. Please use a table in the methods section to outline all your conducted experiments, including key information about these experiments.

**3 Minor Comments**

Line 1 (Title): I suggest incorporating warming impact on changes and removing PGW. A revised title could be: "Impact of warming on rainfall changes in damaging Philippine typhoons using high-resolution convection-permitting models"

Line 49: I did not find the corresponding references of Wang et al. (2014, 2015) in the Reference section. Also, I noticed that the author name does not match the article titled 'Super Clausius-Clapeyron scaling of extreme hourly convective precipitation and its relation to large-scale atmospheric conditions'.

Line 60: "Recent literature describes ..." is better replaced with "Recent studies report ..."

Line 62: "rainfall rates within a 100-km radius" would be more precise if changed to "average rainfall rates within a 100-km radius".

Line 75: "the study hopes to provide additional evidence". It is better to delete 'hopes to'. I believe the article provides such additional evidence.

Line 129: TC inner core region is defined typically using distance like 1 degree, or 100 km or twice of radius of maximum wind. I suggest to defining inner core aslo using 1 degree to see whether results have changed.

Line 166: In terms of Figure 2 (middle panel), BOPHA in the future climate indeed shows weaker rainfall and relative humidity than the current climate during some hours. Do you have any ideas why this occurs?

Lines 166–170: Considering that BOPHA in the future climate indeed shows weaker rainfall and relative humidity, it may not be suitable to simply state that 'Under future climate conditions, accumulated precipitation consistently and markedly increases relative to the current climate'. Try to be more precise in describing the contents in Figure 2.

Line 168: Remove the excessive dot before 'Similarly'.

Line 207: Figure 4 lacks units on the y-axis ('total accumulated rainfall'), and also provide unit information in the Figure 4 caption.

Line 223: The unit should be corrected.

Line 213: Why did you choose to use daily TC rain rate instead of hourly rain rate or 6-hour rain rate as discussed in Figure 6? I suggest using a consistent rain rate metric.

Figure 4: I suggest creating a box plot for total accumulated rainfall over multiple averaging radii, such as 500 km and 100 km. The 500 km averaging radius is typically used to account for the total TC rainfall area. The 100 km averaging radius is used to account for the TC inner core rainfall.

Figure 4: Also lacks units in both the caption and the figure.

Figure 6: 1) Please consider a logarithmic scale for the y-axis. 2) Please label your subfigures properly using a format like a), b), c). 3) Please set your title for the x-axis properly. 4) What does "the black" mean? The last sentence of the figure caption is not finished. 5) The legend colors do not align with the corresponding colors. 6) The x-axis title shows 3-hour rain rate, while the main text shows 6-hour rain rate.

Line 226: The sentence "Figure 6 shows the boxplots of percent changes in mean, 95th, and 99th percentiles of 6-hourly rainfall rates for Haiyan, Bopha, and Mangkhut relative to the current climate scenario" is not accurate. Change it to "Figure 6 shows the boxplots of percent changes in mean, 95th, and 99th percentiles of 6-hourly rainfall rates for Haiyan, Bopha, and Mangkhut of the Pre-industrial and Future simulations relative to the current climate scenario".

Lines 227–231: You compared future simulations with the current climate; however, the comparison between Pre-industrial and current climate is ignored, although it seems that the Pre-industrial shows increased TC rainfall rate (e.g., 99th percentile).

Figure 11: Please show the slope, correlation coefficient, and p-values for both a) and b).

Figure 11: Please show units of rain rate and TC intensity in the caption. Please clarify how you defined 'inner core'—using a 100 km averaging radius, or 2 times the radius of maximum wind, or otherwise.

Figure 11: The legend is too small. I suggest splitting the legend into two parts in Figure 11 b) if there are too many. Also, define short names for model setups which can be consistently referenced throughout the paper, such as referencing Haiyan as HY.

Figure 12 is not well interpreted in the main text. There should be some discussions regarding Preind-Ctrl in your main text.

Figure 12: The units (dBZ) and experiment setups are displayed in the subfigure titles. You can probably improve the figure by clarifying units and experiment setups in proper locations.

Line 317: What does "deep core convection" mean? Or is it deep convection?

Line 332: Add a blank space after (Hendricks et al., 2010).

Line 421: Can you clarify the increase up to 25% at what scenario? And what is the percent increase per degree of warming?

---

## Referee Comment (RC2)

**Revision of "Changes in tropical cyclone-associated precipitation of highly damaging Philippine typhoons using high-resolution PGW simulations and multiple-experiment approach"**

**Overall Summary**

The manuscript by Delfino et al., explores the influence of global warming on three different tropical cyclones (TCs) in a past, present and future climates, imposed by different set-ups contemplating the role of SST, atmospheric temperature and relative humidity at different configurations and resolutions. This is a complete research going from the changes in precipitation rates and other characteristics of the TCs, to addressing possible reasons for the observed super Clausius-Clapeyron behavior in the Philippines. However, I would suggest major revisions to help clarify some aspects of the manuscript.

**Major comments**

1. In the experimental design please be more concise about the prescribed conditions. How are you defining pre-industrial and future conditions? What global warming level are you defining for the future? This sentence in line 110 *"Monthly mean deltas from CMIP6 (2070–2099 minus historical)"* is not clear enough as to what are those years referring to.

2. About the analysis done for the rainfall percentage change summarized in Fig 6. From the text, I understand that the change (%) is done for the difference between (Future – Current) and (Pre-industrial – Current) conditions. This arises an issue to me when comparing to the 7% / K reference. Since it is not clear in the text what is the difference in SSTs between Future and Current times, I can't tell if this change is equivalent to the reference. For instance, if the difference in SSTs between Future and current is 2K, the change in rainfall shown in Fig. 6 corresponds to X% / 2K. The same applies to the pre-industrial. Have they been scaled by the difference between scenarios, or are they absolute differences? Please specify it, in that case that they have not been scaled, the comparison to CC is not directly valid. I advise discussing it in more detail, since this is one of the main results.

3. Is there a reason why the 3Km simulation of BOPHA (Fig1b) has no change in precipitation in the pre-industrial setting? This differs largely from the difference seen in the 5Km simulation.

4. There are a lot of experiments, please include a table describing them.

5. Please revise the figures, many of the labels and text are too small to be read. Also, some are missing the subplot labels (a,b,c...)

**Minor comments:**

1. Disagree with lines 216-217, for Mangkhut the upper tail does increase in frequency, but there is a shift towards decreasing rates in the future (Fig 5).
2. Line 39: Typo. Tropical cyclones (TCs) are a major source
3. Line 46: Add citation to IPCC.
4. Line 47: Define CCS or leave as CC scaling.
5. Line 51: space missing before '. However'.
6. Lines 104-105: Cite CMIP6 and PWG method.
7. Line 110: define years set for historical period.
8. Line 112: '(3) SST, temperature, and humidity (FULL)'. Specify atmospheric temperature.
9. The text in Fig 1 is too small.
10. Line 150: What does CU and NoCU stand for in the experiments?
11. Line 168: Typo, there is a double dot.
12. Line 424: Typo, there is a double dot.
13. Please check the bibliography, some references are missing DOI.
14. References also have different formats in the bibliography.